# Encouraging Tactics with Genetically Modified Probiotics to Improve Immunity for the Prevention of Immune-Related Diseases including Cardio-Metabolic Disorders

**DOI:** 10.3390/biom13010010

**Published:** 2022-12-21

**Authors:** Tomoko Asai, Sayuri Yoshikawa, Yuka Ikeda, Kurumi Taniguchi, Haruka Sawamura, Ai Tsuji, Satoru Matsuda

**Affiliations:** Department of Food Science and Nutrition, Nara Women’s University, Nara 630-8506, Japan

**Keywords:** cardio-metabolic disorders, obesity, diabetes, MAFLD, NAFLD, PI3K, AKT, mTOR, probiotics, genetically modified probiotics

## Abstract

The PI3K/AKT/mTOR signaling pathway may play crucial roles in the pathogenesis of obesity and diabetes mellitus, as well as metabolic syndromes, which could also be risk factors for cardio-metabolic disorders. Consistently, it has been shown that beneficial effects may be convoyed by the modulation of the PI3K/AKT/mTOR pathway against the development of these diseases. Importantly, the PI3K/AKT/mTOR signaling pathway can be modulated by probiotics. Probiotics have a variety of beneficial properties, with the potential of treating specific diseases such as immune-related diseases, which are valuable to human health. In addition, an increasing body of work in the literature emphasized the contribution of genetically modified probiotics. There now seems to be a turning point in the research of probiotics. A better understanding of the interactions between microbiota, lifestyle, and host factors such as genetics and/or epigenetics might lead to a novel therapeutic approach with probiotics for these diseases. This study might provide a theoretical reference for the development of genetically modified probiotics in health products and/or in functional foods for the treatment of cardio-metabolic disorders.

## 1. Introduction

Dyslipidemia has been established as an important factor that is connected with the pathogenesis of several diseases including atherosclerosis, hypertension, cardiovascular disease, obesity, and diabetes mellitus. Therefore, cardio-metabolic disorders are a set of pathophysiological conditions resulting from a disturbance in hepatic lipid metabolism, which may occupy a position at the crossroads of cardiovascular disease and metabolic disorder [1]. The risk factors of cardio-metabolic disorders may also enhance the morbidity rate for several diseases such as obesity, metabolic syndrome, hypertension, type 2 diabetes (T2D), ischemic heart disease, and stroke [2]. In particular, obesity is now prevalent worldwide following an increase in comorbidities with various metabolic dysfunction including the T2D, metabolic-dysfunction-associated fatty liver disease (MAFLD), and cardio-metabolic disorders [3] (Figure 1). Here, the term “MAFLD” is used instead of the previous term NAFLD (non-alcoholic fatty liver disease), because MAFLD has been suggested as a more appropriate overarching term [4]. Since cardio-metabolic disorders must have existed before they became well-known worldwide, it is crucial for the prevention of diseases to be informed of some altered tactics, as well as treatment arrangements, to regulate the prevalence [5]. Among them, certain types of diet as good therapy for cardio-metabolic disorders could provide beneficial effects for the risk factors. In fact, there is an increasing interest for many researchers in the association between dietary glycemic index and the risk factors of cardio-metabolic disorders. MAFLD is one of the most common liver diseases in the world, in which other cardio-metabolic disorders in patients with MAFLD are significantly prevalent [6]. Interestingly, the associations of metabolic disorders with MAFLD have been greater in non-obese than in obese patients with diabetes mellitus [7]. A chronic pro-inflammatory condition in obesity patients could trigger and/or promote cardio-metabolic disorders [8]. Chronic inflammation is also a risk factor for several conspicuous health concerns related to cardio-metabolic disorders [9]. In addition, lifestyle and/or epigenetic factors may be also involved in the development of MAFLD and cardio-metabolic disorders [10].

The potential therapeutic approaches for the improvement of various diseases might target gut microbiota and/or their metabolites by following the alteration of pathological pathways in diseases [11]. One of these approaches may contain the utilization of probiotics. The human gut microbiota and/or their metabolites have also become a potential therapeutic target for the development of interventions for the prevention of cardio-metabolic disorders. Short-chain fatty acids (SCFAs) are characterized by containing fewer than six carbons, such as acetate, propionate, and butyrate, which are mainly produced by the gut microbiota as fermentation products [12]. In experimental models, one of the SCFAs, propionate, could considerably decrease appetite and/or food intake via the effect of gut–brain axis in animals [13], suggesting that a SCFA-rich diet might be significant for managing obesity and/or associated diseases, as well as several cardiovascular diseases. In fact, the butyrate could attenuate the angiotensin-II effects [14]. Probiotic fermented milk might be able to lower the incidence rate of hypertension. In addition, gut microbiota could affect various human physiological activities in some biological characteristics such as nutrient synthesis and/or immune modulation [15]. Both scientists and clinicians have an extreme interest in the development of probiotics to increase the therapeutic potential. Meanwhile, engineered probiotics have been actually developed as potential therapeutics for a variety of inflammation-related diseases including MAFLD and diabetes [16]. In the present perspective, we would like to discuss the utilization of probiotics including genetically modified probiotics, paying attention to how these probiotics could be employed as a therapeutic strategy against the pathogenesis of cardio-metabolic disorders.

## 2. PI3K/AKT and AMPK Pathway Involved in Obesity, MAFLD, and Diabetes

Phosphoinositide 3-kinase (PI3K) and AKT, a serine/threonine protein kinase also known as protein kinase B, are key components with the mechanistic target of rapamycin (mTOR) for signaling network [17], possibly related to the pathogenesis of cardio-metabolic disorders. The mTOR is also a serine/threonine kinase located downstream of the PI3K/AKT signaling pathway [18]. (Figure 1) Aberrant activation of the PI3K/AKT/mTOR network may contribute to pathological conditions including T2D, MAFLD, and cancers. Therefore, modulating the components in the PI3K/AKT/mTOR pathway has been proposed as a potential therapeutic option for preventing and/or treating diverse types of host disorders, whose chronic complications are substantial burdens in social communities [19]. For example, insulin-facilitated PI3K/AKT signaling is important for the regulation of reproductive dysfunction and/or metabolic abnormalities [20]. Similarly, reduced insulin activity or insulin resistance could result in impaired PI3K/AKT signaling [20], which controls downstream factors such as mTOR and/or GSK-3β [21]. In addition, the PI3K/AKT-associated suppression of the mTOR pathway has been identified as a mediator for the activation of autophagy following hypoxia [22], which appears to be beneficial for resisting cardiac hypertrophy. Furthermore, the protective effects of resveratrol, such as the reduction in apoptosis in kidney cells and/or the attenuation of high-glucose-induced oxidative stress, might be exerted through the modulation of the PI3K/AKT/mTOR pathway in patients with diabetic nephropathy [23]. Inhibition of the progression of diabetic cardiomyopathy could be achieved via stimulating the PI3K/AKT signaling pathway [24]. In addition, anti-inflammatory or antioxidant features of the PI3K/AKT signaling pathway might be advantageous for diabetic cardiomyopathy [24]. Then, activation of the PI3K/AKT pathway might be a possible mechanism that contributes to the protection of diabetic myocardium [25]. In addition, decreased apoptosis in cardiomyocytes might be also linked to the activation of PI3K/AKT/mTOR signaling [26]. Hence, understanding the practical role of the cardiac apoptosis in the pathogenesis of diabetic cardiomyopathy may be a brilliant way to improve more targeted treatments [26]. In addition, obesity-related acute pancreatitis is related to the PI3K/AKT/mTOR signaling pathway [27]. Aberrant activation of the PI3K/AKT/mTOR signaling has been reported to be associated with a wide variety of human diseases including MAFLD [28]. Interestingly, blockade of ghrelin receptor could take care of MAFLD, possibly via the hypothalamic PI3K/AKT/mTOR signaling, to improve insulin resistance [29]. In addition, inhibition of the PI3K/AKT/mTOR pathway could slow down the senescence of hepatocytes, then improve MAFLD [30].

Glucose uptake through GLUT4 might be stimulated by anthocyanin via the upregulation of the PI3K/AKT and adenosine monophosphate (AMP)-activated protein kinase (AMPK) signaling pathways [31]. AMPK is a widely existing protein kinase activated by AMP, which is a central regulator of cellular energy balance. It is well-known that AMPK plays an important role in fatty acid metabolism through the fatty acid biosynthetic pathway. In general, AMPK is located in liver and/or in muscle fibers, where AMPK might work as an indicator or a good marker for cellular energy balance. AMPK is also a metabolic sensor molecule regulating cellular energy metabolism, which might be also involved in anti-lipid metabolism [32]. As a consequence, inhibition of AMPK may result in decreased fatty acid oxidation and increased fat accumulation, and vice versa [33]. Suppressing the AMPK, therefore, could decrease the liver fatty acid oxidation in patients of obesity. In general, PTEN/PI3K and AMPK signaling pathways are modulated by classical antidiabetic drugs [34]. AMPK activators, metformin, have been shown to prevent the development of hepatic steatosis [35]. Consequently, AMPK is closely associated with the content of liver lipid and/or insulin resistance [36]. An experimental study shows that protein expression levels of AMPK in high-fat-diet-induced MAFLD is considerably lower than that of the normal control group [37], which may suggest that AMPK is actually involved in the pathogenesis of MAFLD.

## 3. Probiotics for the Modulation of PI3K/AKT and AMPK Pathway

Probiotics, as naturally occurring bacteria that are detected in human and/or animal intestine, revealed their capability against the development of various diseases via the alteration of intracellular signaling pathways within target cells. Importantly, the PI3K/AKT signaling pathway could be successfully modulated by probiotics [38], which might offer important therapeutic options by targeting cellular apoptosis, survival, and/or protection. For example, modulation of the PI3K/AKT pathway has a considerable hepatoprotective mechanism, which could be achieved after *Akkermansia muciniphila* (*A. muciniphila*) administration [39]. *Lactiplantibacillus plantarum* (*L. plantarum*) treatment could also enhance the expression of PI3K/AKT in the liver, which could prevent high-fat-diet-induced glucose tolerance and/or hyperglycemia to improve type 2 diabetes mellitus [40,41]. Altered gut microbiota components might influence liver glycogen and/or muscle glycogen by elevating the mRNA expression of PI3K and/or AKT in the liver through the modulation of favorable bacteria such as *Lactobacillus* sp. [42]. *Lactobacillus rhamnosus* (*L. rhamnosus*) could prevent cytokine-induced cellular apoptosis in gut epithelial cells by up-regulating the PI3K/AKT signaling pathway [43]. Probiotic supplementation is a potential way to protect against high-fat-diet-induced and radiation-induced liver damage, possibly via the alteration of the PI3K/AKT signaling pathway [44]. *Bifidobacterium animalis* subsp. *lactis* (*B. lactis)* strain could retard an apoptosis of the colonic epithelial cells by up-regulating the PI3K/AKT signaling pathway [45]. Interestingly, *B. lactis* combined with *L. plantarum* could regulate the growth of malignant glioma by suppressing the PI3K/AKT pathway in mice [46].

As mentioned above, AMPK is a significant molecule involved in regulating biological energy metabolism [47] and/or in the regulation of aging, which could activate downstream molecules such as sirtuin1 (SIRT1) to inhibit inflammation and/or oxidative stresses [48]. *L. plantarum* could activate AMPK for reducing the production of ROS and inflammation, which might contribute to inhibiting atherosclerosis [49]. On the other hand, a gut-microbiota-dependent metabolite could bring various cardiovascular diseases [49]. It is well-known that alteration of immune cells balance might be related to the inflammatory response through the AMPK and/or PI3K/AKT signaling pathway (Figure 2). In addition, *L. plantarum* supplementation could adjust the AMPK signaling pathway, which could amend the insulin signaling pathway, fatty acid biosynthesis, fatty acid metabolism, and the glucagon signaling pathway [50]. *A. muciniphila* could successfully enhance the AMPK activation for the improvement of inflammatory responses and/or the restoration of intestinal barrier function [51]. A blend of *Streptococcus thermophilus* and *Lactococcus lactis* (*L. lactis*) could also protect the liver by the activation of AMPK-mediated signaling, thereby promoting lipid oxidation [52]. *Bacillus licheniformis* could activate AMPK in hepatocytes for the regulation of gene expression associated with lipid metabolism [53]. *L. rhamnosus* could also activate the AMPK pathway, which may have positive effects on hyperlipidemia by lowering the serum lipid concentration and improving the lipid profile [54]. Interestingly, *L. rhamnosus* may increase the longevity-related bacteria such as *Bifidobacterium*, *Lactobacillus*, and *A. muciniphila* in the gut, which could also activate the AMPK pathway and contribute to the pathogenesis of age-related diseases [55].

## 4. Possible Mechanism behind the Beneficial Effects of Probiotics on Obesity and Diabetes

Living microorganisms may have a valuable effect, with probiotics being beneficial for gut microbiota in human health [56]. Therefore, gut microbiota and/or probiotics could be an important mediator for a communication between diet and host health [57]. Microbial metabolites can influence host gene expression through the epigenetic mechanisms [58]. In addition, the mechanisms affecting the expression of PI3K/AKT and AMPK signaling molecules could be associated with the alteration of cardio-metabolic parameters [59]. Microbial metabolites such as SCFAs could influence the epigenetic programming by inhibiting histone deacetylase (HDAC) enzymatic activity [60], which promotes de-condensation and relaxation of chromatin and increases the chromatin accessibility to various transcription factors [61]. *Faecalibacterium prausnitzii* is one of the most abundant anaerobic bacteria in the gut of healthy individuals that can produce SCFAs such as butyrate. A strain of *Lactobacillus paracasei* was initially isolated from the feces of an elderly Italian person, which could prevent and/or improve the pathology of type 2 diabetes mellitus by reduction in inflammation via the alteration of the PI3K/AKT pathway and the production of gut-microbiota-derived metabolites such as SCFAs [62] (Figure 2). Acetate, butyrate, and vitamins all play important roles in epigenetic regulation, which are mainly byproducts of the gut microbiota. After dietary intervention with *L. rhamnosus*, the concentration of acetic acid, propionic acid, and butyric acid might be significantly raised [63]. Therefore, it might be a potentially valuable choice to employ *L. rhamnosus* for acquiring epigenetical effects in probiotics. Obesity is a complex pathology with a multifactorial pathogenesis linked with lifestyle and epigenetic factors [64]. *L. rhamnosus* supplementation use as a probiotic could contribute to a significant decrease in plasma triglycerides, low density lipoprotein (LDL)-cholesterol, insulin, and homeostatic model assessment for insulin resistance (HOMA-IR) [65]. In addition, *L. rhamnosus* could bring improvement in the profile of total cholesterol, LDL, high density lipoprotein (HDL), triglycerides, and weight gain [66]. *L. rhamnosus* could activate the AMPK pathway, and reduce the gene expression of peroxisome proliferator-activated receptor (PPAR) [63]. In addition, *L. rhamnosus* could down-regulate the expression of genes related to adipogenesis and/or lipogenesis in high-fat-diet-fed obese mice [67].

One of the SCFAs, butyrate, a major end product of the bacterial fermentation of indigestible carbohydrates, might have positive effects on body weight control and insulin sensitivity [68]. Sodium butyrate could attenuate the obesity-induced insulin resistance, fatty liver, and intestinal dysfunction [69]. Butyrate has corrected hyperinsulinemia, lowered plasma leptin levels, and attenuated adipose tissue inflammation without affecting gut microbiota composition [70]. Sodium butyrate is a short-chain fatty acid with HDAC inhibition activity, which could epigenetically promote beta-cell development, proliferation, and function, as well as improve glucose homeostasis [71]. Many commensal gut bacteria such as *Lactobacillus* and *Bifidobacterium* species are important in folic acid production [72], which might be involved in methylation changes as the promoters for regulating transcription activity [73]. Understanding the precise and specific epigenetic changes in host genes associated with a cardio-metabolic disorder by certain microbiota could be important in the development of novel therapies against the cardio-metabolic disorder.

## 5. Genetically Modified Probiotics with *Lactococci lactis*

*Lactococci* and *Lactobacilli* have been broadly studied and frequently manipulated for the progress of engineered probiotics [74]. In particular, *L. lactis* has been used as a food-grade and endotoxin-free genetically engineered vector for protein expression [75]. *L. lactis* is a Gram-positive bacterium commonly used in the production of dairy foodstuffs. In addition, this bacterium can synthesize bacteriocins [76], which can inhibit the growth of unwanted microorganisms for preserving the hygienic quality of the foodstuffs [77]. For example, in the usage of applied probiotics, *L. lactis* strains expressing murine IL-10 or IL-35 are considerably therapeutic in the dextran sodium sulfate (DSS)-induced colitis in a mouse model [78,79]. *L. lactis* expressing IL-35 could also protect against rheumatoid arthritis in mice [80]. It is well-known that both inflammatory colitis and rheumatoid arthritis are immune-related diseases. Accordingly, microbiota could also participate in host immune function for the inhibition of disease development. In fact, the human gut microbiota has now become a potential therapeutic target in the development of novel cardio-metabolic agents [11] (Figure 2). Interestingly, expression of IL-10 in combination with glutamic acid decarboxylase (GAD) by *L. lactis* has improved functional β-cell mass and hyperglycemia without any diabetic symptoms in the non-obese diabetic mouse model [81]. Heat shock proteins including *Mycobacterium* HSP65, when expressed in *L. lactis*, could also be protective against DSS-induced colitis [82]. In addition, orally administered recombinant *L. lactis* engineered to express HSP65 might be an effective therapeutic method in preventing diabetes mellitus type 1 [83]. Exendin-4 secreted by the engineered *L. lactis* is a glucagon-like protein-1 (GLP-1) receptor agonist that is thought a good therapeutic peptide for type 2 diabetes, which might theoretically serve as a novel strategy for oral treatment of diabetes [84]. Exendin-4 could activate the PI3K/AKT signal pathway, which enhances the glucose-dependent insulin secretion. In addition, the administration of the engineered *Lactococcus*-expressing Ling Zhi-8 may be a promising treatment for improving MAFLD [85]. Ling Zhi-8 is an immunomodulatory protein isolated from the medicinal mushroom known as Ling Zhi [86,87]. It has been recognized that Ling Zhi-8 owns a broad range of properties such as anti-inflammatory actions [86,87]. Probiotics with various *Lactococci* and *Lactobacilli* strains have also been engineered to address the induction of superoxide dismutase expression for reducing reactive oxygen species (ROS) and/or various oxidative stresses [88]. Human angiotensin-converting enzyme 2 (ACE2) expressed in *Lactobacillus casei* could diminish retinopathy symptoms in diabetic retinopathy mouse models [89,90]. It is well-known that ACE2 is also linked to decreased inflammation and/or oxidative stresses. Elafin is an endopeptidase that prevents elastase-mediated tissue proteolysis associated with inflammatory bowel disease (IBD) [91]. Recombinant lactic acid bacteria combined with delivering the elafin could decrease inflammation in chronic IBD mouse models [92].

In these ways, engineered probiotics are now demonstrating an effective delivery method that could prevent inflammatory diseases [81,93]. The results of these studies further support the rational development of novel probiotics for the treatment of cardio-metabolic disorders. In particular, the pivotal importance of those anti-inflammatory molecules identified in animal models and clinical studies will provide insight into the development of novel therapeutic targets for use in genetically engineered probiotics [94].

## 6. Future Perspectives

The mechanism of probiotics could be, in part, concluded as an alteration or recovery of microbiota balance. Helpfully, engineered probiotics could serve as optimum vectors to produce beneficial molecules targeting specific factors, cells, organs, and/or even pathogens [95]. The development of gene editing tools, such as clustered regularly interspaced palindromic repeat (CRISPR)/CRISPR-associated proteins (Cas), (CRISPR/Cas), is a milestone event in engineering. By means of the genome editing and/or recombineering, probiotics now possess a variety of potentially beneficial properties, which could treat specific diseases and contribute to human health and/or quality of life (QOL). This might be also a turning point in the research of probiotics. CRISPR might offer indispensable support for the emergence of the next generation of probiotics [96]. The CRISPR/Cas9 system has been initially used for selection against cells lacking the desired chromosomal modification [97]. In addition, a single-strand DNA engineering method for gene mutation in *L. lactis* has been further defined [98]. In combination with the CRISPR/Cas9 selection, the single-strand DNA engineering could decrease the construction time with *L. lactis* from several weeks to 3 days [96]. Engineered *Lactobacilli* are being developed as targeted therapies against a wide range of diseases such as IBD and viral/bacterial infections [99]. In addition, a method was developed to perform genome editing in *Lactobacillus reuteri* [100]. CRISPR tools could be used to enhance therapeutic effects of lactic acid bacteria [101]. In *Lactobacillus plantarum*, CRISPR/Cas9-mediated genome editing has been also presented [102]. Furthermore, CRISPR–Cas9 tools have already been developed for other bacterial species including *Bacillus subtilis* [103]. In these ways, the CRISPR/Cas9 genome editing tools would be employed in applications such as metabolic engineering [104] for the treatment of cardio-metabolic disorders. The CRISPR technology is consistent, but sometimes, an “off-target” phenomenon would occur [105]. Meanwhile, it still takes a long time before putting it into use in clinical treatment. CRISPR tools have provided sensational strategies to progress the development of novel therapies. Therefore, more studies are necessary immediately.

## Figures and Tables

**Figure 1 biomolecules-13-00010-f001:**
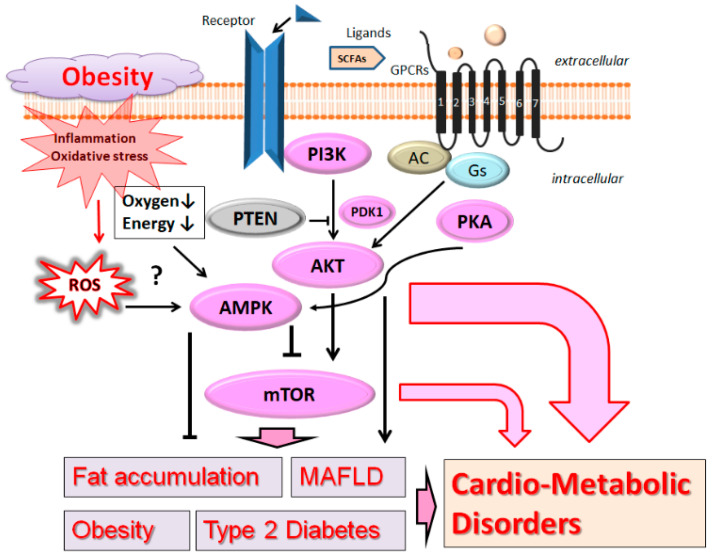
A hypothetical schematic image and overview of the pathogenesis of cardio-metabolic disorders. The PI3K/AKT/AMPK/mTOR signaling pathway might intricately contribute to the pathogenesis of obesity, type 2 diabetes mellitus, metabolic-associated fatty liver disease (MAFLD), which are all related to the pathogenesis of cardio-metabolic disorders. Arrowhead indicates stimulation, whereas hammerhead shows inhibition. Note that several important activities such as inflammatory-related reactions have been omitted for clarity. Abbreviation: MAFLD, metabolic-associated fatty liver disease; AMPK, adenosine monophosphate-activated protein kinase; mTOR, mammalian/mechanistic target of rapamycin; PI3K, phosphoinositide-3 kinase; PKA, protein kinase A; PTEN, phosphatase and tensin homologue deleted on chromosome 10; ROS, reactive oxygen species. (Numbers in black boxes: G protein-coupled receptors (GPCRs) are commonly seven-(pass)-transmembrane domain receptors).

**Figure 2 biomolecules-13-00010-f002:**
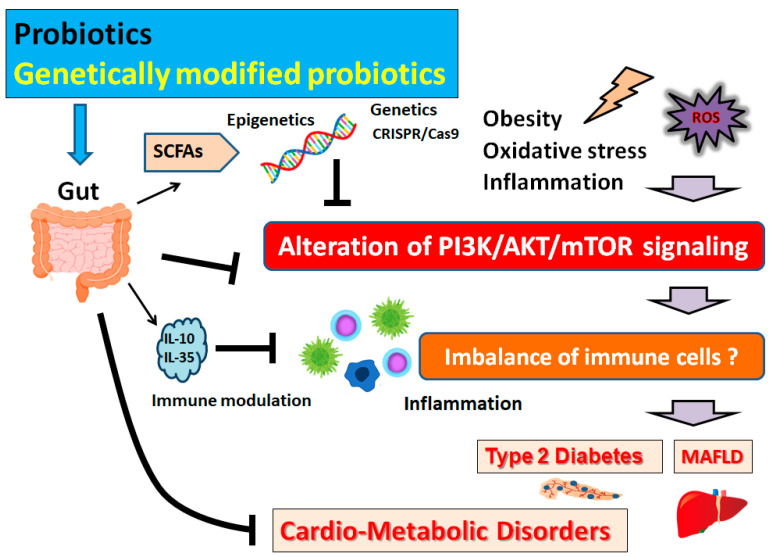
Schematic representation of the inhibition of the pathogenesis of cardio-metabolic disorders. Some types of probiotics and/or genetically modified probiotics could contribute to the alteration of gut microbial community for playing valuable roles in inhibiting the PI3K/AKT/mTOR signaling pathway in part via the epigenetics regulation with SCFAs. Examples of certain beneficial microbial species with several effects on anti-cancer immune responses are shown on the left side. Arrowhead indicates stimulation whereas hammerhead shows inhibition. Note that several important activities such as cytokine induction and/or inflammatory reactions have been omitted for clarity. Abbreviation: MAFLD, metabolic-associated fatty liver disease; mTOR, mammalian/mechanistic target of rapamycin; PI3K, phosphoinositide-3 kinase; ROS, reactive oxygen species; SCFAs, short-chain fatty acids.

## Data Availability

Not applicable.

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
