# Peer review of "Encouraging Tactics with Genetically Modified Probiotics to Improve Immunity for the Prevention of Immune-Related Diseases including Cardio-Metabolic Disorders"

_biomolecules, 2022, doi:10.3390/biom13010010_

Round 1

Reviewer 1 Report

The authors described this perspective on GM probiotics and their relation with the mitigation of cardiometabolic disorders through the regulation signaling pathways.

There are some specific comments below.

Line 13: replace ‘those’ with ‘these’

Line 19: replace ‘those’ with ‘these’

Line 28: remove ‘.’

Line 39: put “:” after IBD

Line 51: put “:” after PPAR

Line 85-86: Rearrange the sentence

Line 109: replace ‘those’ with ‘these’

Line 122-123: Rewrite the sentence

Line 140: include ‘of’ before the PI3K/AKT/mTOR

Line 151-152: Rearrange the sentence

Line 167: remove ‘.’ Akkermansia.

Line 193: change capital letter to small ‘Lactis’

Line 2014-215: Rewrite the sentence

Line 227: remove “via the” before production of gut microbiota

Line 261: change capital letter to small ‘Lactis’

Line 304-305: Rewrite the sentence

Line 308-309: Rewrite and complete the sentence

Line 324-327: Suggested deleting the content, as the article mainly describe cardiometabolic disorders, not need to discuss AMR.

Author Response

Reviewer 1

The authors described this perspective on GM probiotics and their relation with the mitigation of cardiometabolic disorders through the regulation signaling pathways.

There are some specific comments below.

 Thank you so much for the intellectual suggestions.

Line 13: replace ‘those’ with ‘these’

done

Line 19: replace ‘those’ with ‘these’

done

Line 28: remove ‘.’

done

Line 39: put “:” after IBD

done

Line 51: put “:” after PPAR

done

Line 85-86: Rearrange the sentence

Rearrangement has been done.

Line 109: replace ‘those’ with ‘these’

done

Line 122-123: Rewrite the sentence

The sentence has been rewritten.

Line 140: include ‘of’ before the PI3K/AKT/mTOR

done

Line 151-152: Rearrange the sentence

The sentence has been improved.

Line 167: remove ‘.’ Akkermansia.

done

Line 193: change capital letter to small ‘Lactis’

done

Line 2014-215: Rewrite the sentence

The sentence has been improved.

Line 227: remove “via the” before production of gut microbiota

done

Line 261: change capital letter to small ‘Lactis’

done

Line 304-305: Rewrite the sentence

The sentence has been rewritten.

Line 308-309: Rewrite and complete the sentence

The sentence has been improved.

Line 324-327: Suggested deleting the content, as the article mainly describe cardiometabolic disorders, not need to discuss AMR.

Agreeing with this suggestion, we have deleted the content. Thank you so much.

Reviewer 2 Report

The manuscript by Asai et al. summarizes the role of genetically probiotics in the treatment of cardio-metabolic disorders. 

The topic is novel, but unfortunately, the study lack at many aspects, such as- language editing, presentation of concept and study design, improper use references, poor flow in the text. Some of the detailed comments are:

1. Paragraph 1, mainly discusses MAFLD not the cardio-metabolic disorders. Authors should first define their meaning of " cardio-metabolic disorders"

2. Figure 1 does not really fit into the manuscript or needs to be more specific to the topic.

3. Substantial language editing is needed. For example:

-Line 59-61 (The risk.....stroke), authors say risk factors for cardio metabolic disorders may also enhance the possibilities of obesity....stroke. Is cardio metabolic disorders are apart from the disorders listed or the same?

-lines 66-69 are hard to comprehend. Eminence might not be the correct word to be used in the sentence.

-"Diet as a go therapy" sounds non-scientific, however the sentence itself makes little sense to me.

-Sentence (line 95-97) is hard to comprehend. 

Almost every sentence needs to be paraphrased by professional English editors.

4. Inappropriate use of references: For example- 

-For the lines (93-95), Short chain.......Product [12], reference 12 doesn't look appropriate.

-Reference 14 in the line 99-101, is probably an incorrect citation.

-Most of the article cited are review articles, which is very confusing and probably false information.

5- Section 2 and 3 could be just one section, emphasizing on the later one.

6. Authors could have included a table summarizing the section 5. It is hard to follow the text. Table could list the probiotics, their genetic modification, therapeutic/nutrition importance.

Author Response

Reviewer2

The manuscript by Asai et al. summarizes the role of genetically probiotics in the treatment of cardio-metabolic disorders. 

The topic is novel, but unfortunately, the study lack at many aspects, such as- language editing, presentation of concept and study design, improper use references, poor flow in the text. Some of the detailed comments are:

  1. Paragraph 1, mainly discusses MAFLD not the cardio-metabolic disorders. Authors should first define their meaning of " cardio-metabolic disorders"

According to this indication, we have improved the text of paragraph 1.

  1. Figure 1 does not really fit into the manuscript or needs to be more specific to the topic.

Figure 1 has been amended to be specific to the topic.

  1. Substantial language editing is needed. For example:

-Line 59-61 (The risk.....stroke), authors say risk factors for cardio metabolic disorders may also enhance the possibilities of obesity....stroke. Is cardio metabolic disorders are apart from the disorders listed or the same?

We think it is same one, or it could not be distinguished. They might be interconnected and mutually.

-lines 66-69 are hard to comprehend. Eminence might not be the correct word to be used in the sentence.

The word “eminence” has been replaced with the word “well-known”.

-"Diet as a go therapy" sounds non-scientific, however the sentence itself makes little sense to me.

We are thinking about the more appropriate word for the meaning. Do you have any suggestion?

-Sentence (line 95-97) is hard to comprehend. 

The sentence has been improved.

Almost every sentence needs to be paraphrased by professional English editors.

According to the suggestion, we have gone over the text/abstract and amended typos and grammatical errors as much as possible with a help of native English-speaker to improve the manuscript more helpful to the readers.

  1. Inappropriate use of references: For example- 

-For the lines (93-95), Short chain.......Product [12], reference 12 doesn't look appropriate.

Reference 12 has been replaced.

-Reference 14 in the line 99-101, is probably an incorrect citation.

Reference 14 has been also altered.

-Most of the article cited are review articles, which is very confusing and probably false information.

Several review articles cited such as reference 25, 36, and 59 have been replaced with the related original articles.

5- Section 2 and 3 could be just one section, emphasizing on the later one.

According to this suggestion, we have improved the expression of the section 2, and 3 paragraphs in addition to that of section 1 in order to interconnect logically with the first 3 paragraphs, which might be beneficial to the readers for understanding the relationship between gut microbiota and cardio-metabolic disorders.

  1. Authors could have included a table summarizing the section 5. It is hard to follow the text. Table could list the probiotics, their genetic modification, therapeutic/nutrition importance.

Thank you so much. That is a good idea. However, it would be too tough for us to make the nice table at this stage of the manuscript-revision. We would like to include it on the manuscript in the future chance of writing.

Reviewer 3 Report

This article aims to review on the potential applications of genetically modified probiotics in the treatment of cardio-metabolic disorders as well as the mechanisms underlying the beneficial effects of probiotics. This topic sounds interesting, however, there are some concerns as follows:

1. Although the title is Encouraging tactics with genetically modified probiotics for the treatment of cardio-metabolic disorders, the authors just talked a little bit about the relationship between probiotics and cardio-metabolic disorders. Instead, they talked much about the potential applications of probiotics in the treatment of obesity, diabetes mellitus and metabolic syndrome. I thus suggest that the authors either revise the title corresponding to the content of this article or revise the content to focus on cardio-metabolic disorders.

2. It seems that the first and second paragraphs are not logically coherent. I think the authors may insert a paragraph talking about the relationship between gut microbiota and cardio-metabolic disorders before talking about the potential therapeutic values of probiotics.

3.  Line 151-153, the authors mentioned that By suppressing the AMPK, therefore, the gut microbiota could decrease the liver fatty acid oxidation in patients of obesity. The authors however did not talk about how gut microbiota suppress the AMPK.

Author Response

Reviewer3

This article aims to review on the potential applications of genetically modified probiotics in the treatment of cardio-metabolic disorders as well as the mechanisms underlying the beneficial effects of probiotics. This topic sounds interesting, however, there are some concerns as follows:

  1. Although the title is ‘Encouraging tactics with genetically modified probiotics for the treatment of cardio-metabolic disorders’, the authors just talked a little bit about the relationship between probiotics and cardio-metabolic disorders’. Instead, they talked much about the potential applications of probiotics in the treatment of obesity, diabetes mellitus and metabolic syndrome. I thus suggest that the authors either revise the title corresponding to the content of this article or revise the content to focus on cardio-metabolic disorders.

Thank you so much for the good suggestion. We have altered the title corresponding to the content.

  1. It seems that the first and second paragraphs are not logically coherent. I think the authors may insert a paragraph talking about the relationship between gut microbiota and cardio-metabolic disorders before talking about the potential therapeutic values of probiotics.

To interconnect logically with the first two paragraphs, we have improved the expression both of the paragraphs, which might be beneficial to the readers for understanding the relationship between gut microbiota and cardio-metabolic disorders.

  1. Line 151-153, the authors mentioned that ‘By suppressing the AMPK, therefore, the gut microbiota could decrease the liver fatty acid oxidation in patients of obesity’. The authors however did not talk about how gut microbiota suppress the AMPK’.

According to the indication, the sentence has been altered as “By suppressing the AMPK, therefore, something to inhibit the AMPK could decrease the liver fatty acid oxidation in patients of obesity”.

Round 2

Reviewer 2 Report

Authors have improved the manuscript. However, some minor points are still there to be focused on. 

My further comments are:

1- Section 5 should discuss 'genetically modified probiotics and their role in preventing/treating cardio metabolic disorders'. However, the authors should avoid generalizing the topic to other diseases, such as IBD. It is the key section of the article, which needs to be focused, otherwise the title would be unrelated.  

2- Reference 59, in the line 234-235, doesn't look appropriate, as it has no relevance to probiotics. The study mainly discusses effects of metformin on aortic valves.

3- As section 4 does not specifically discuss epigenetic pathways, the heading should be "Mechanism behind beneficial effect of probiotics in obesity and diabetes" However the second part of the section talks.

4- As per my previous comments, adding a table summarizing the genetically modified (unfortunately very few) probiotics, their role in metabolism, and therapeutic implication.

Author Response

For reviewer 2

1- Section 5 should discuss 'genetically modified probiotics and their role in preventing/treating cardio metabolic disorders'. However, the authors should avoid generalizing the topic to other diseases, such as IBD. It is the key section of the article, which needs to be focused, otherwise the title would be unrelated.  

At present, there is very few data which shows the clear relationship or interaction between genetically modified probiotics and cardio metabolic disorders. So, we have altered the manuscript title to “Encouraging tactics with genetically modified probiotics to improve immunity for the prevention of immune-related diseases including cardio-metabolic disorders”. In addition, we have added some explanation for mentioning with the IBD in Section 5.

2- Reference 59, in the line 234-235, doesn't look appropriate, as it has no relevance to probiotics. The study mainly discusses effects of metformin on aortic valves.

 Reference 59 has been replaced with the other one so as to make sense.

3- As section 4 does not specifically discuss epigenetic pathways, the heading should be "Mechanism behind beneficial effect of probiotics in obesity and diabetes" However the second part of the section talks.

Yes, absolutely. Thank you so much. The heading of section 4 has been improved as "Possible mechanism behind the beneficial effects of probiotics in obesity and diabetes".

4- As per my previous comments, adding a table summarizing the genetically modified (unfortunately very few) probiotics, their role in metabolism, and therapeutic implication.

Again, there is very few data about them. In particular, in the field of cardio-metabolic disorders. Please let me show them on the next manuscript for the similar issue in the future.